# Limb Remote Ischemic Conditioning Promotes Neurogenesis after Cerebral Ischemia by Modulating miR-449b/Notch1 Pathway in Mice

**DOI:** 10.3390/biom12081137

**Published:** 2022-08-18

**Authors:** Sijie Li, Yong Yang, Ning Li, Haiyan Li, Jiali Xu, Wenbo Zhao, Xiaojie Wang, Linqing Ma, Chen Gao, Yuchuan Ding, Xunming Ji, Changhong Ren

**Affiliations:** 1Beijing Key Laboratory of Hypoxia Translational Medicine, Xuanwu Hospital, Center of Stroke, Beijing Institute of Brain Disorder, Capital Medical University, Beijing 100053, China; 2Emergency Department, Xuan Wu Hospital, Capital Medical University, Beijing 100053, China; 3School of Traditional Chinese Medicine, Beijing University of Chines Medicine, Beijing 100029, China; 4Department of Neurology, Shenzhen Qianhai Shekou Free Trade Zone Hospital, Shenzhen 518054, China; 5Department of Neurology, The People’s Hospital of Suzhou New District, Suzhou 215129, China; 6Department of Neurosurgery, Wayne State University School of Medicine, Detroit, MI 48201, USA

**Keywords:** ischemic stroke, neurogenesis, limb remote ischemic conditioning, miR-449b, Notch1

## Abstract

Neurogenesis plays an important role in the prognosis of stroke patients and is known to be promoted by the activation of the Notch1 signaling pathway. Studies on the airway epithelium have shown that miR-449b represses the Notch pathway. The study aimed to investigate whether limb remote ischemic conditioning (LRIC) was able to promote neurogenesis in cerebral ischemic mice, and to investigate the role of the miR-449b/Notch1 pathway in LRIC-induced neuroprotection. Male C57BL/6 mice (22–25 g) were subjected to transient middle cerebral artery occlusion (MCAO), and LRIC was performed in the bilateral lower limbs immediately after MCA occlusion. Immunofluorescence staining was performed to assess neurogenesis. The cell line NE-4C was used to elucidate the proliferation of neuronal stem cells in 8% O_2_. After LRIC treatment on day 28, mice recovered neurological function. Neuronal precursor proliferation was enhanced in the SVZ, and neuronal precursor migration was enhanced in the basal ganglia on day 7. LRIC promoted the improvement of neurological function in mice on day 28, promoted neuronal precursor proliferation in the SVZ, and enhanced neuronal precursor migration in the basal ganglia on day 7. The neurological function score was negatively correlated with the number of BrdU-positive/DCX-positive cells in the SVZ and striatum. LRIC promoted activated Notch1 protein expression in the SVZ and substantially downregulated miR-449b levels in the SVZ and plasma. In vitro, miR-449b was found to target Notch1. Lentivirus-mediated miR-449b knockdown increased Notch1 levels in NE-4C cells and increased proliferation in the cells. The effects of miR-449b inhibition on neurogenesis were ablated by the application of Notch1 shRNA. Our study showed that LRIC promoted the proliferation and migration of neural stem cells after MCAO, and these effects were modulated by the miR-449b/Notch1 pathway.

## 1. Introduction

Ischemic stroke is the second leading cause of death worldwide [1]. Currently, methods proven effective for the treatment of acute ischemic stroke include vascular recanalization and neuroprotection. However, recanalization is not associated with a good prognosis [2,3,4]. Therefore, the promotion of neurological function recovery after ischemic stroke and further improvement of prognosis is of utmost importance [5].

After ischemic stroke, neural stem cells in the brain tissue proliferate to give rise to new cells which can migrate to ischemic areas and differentiate into mature neurons [6]. This neurogenesis mechanism plays an important role in the prognosis of stroke patients [7]. Reports have shown that inhibition of neurogenesis can significantly inhibit nerve function recovery in an ischemic stroke model [8]. Transplantation of neural stem cells or administration of erythropoietin can promote neurogenesis and improve neurological function [9,10]. Therefore, promoting neurogenesis and improving neurological function are considered important strategies for the treatment of ischemic stroke.

Although endogenous regeneration was found to be capable of replacing lost neurons, it was quite limited [6]. In recent years, accumulating evidence has focused on non-drug neuroprotection and recovery of neurological function in stroke therapy [11,12]. The concept of ischemic conditioning refers to the initiation of endogenous protective mechanisms against ischemic injury of multiple organs [13]. Zhao et al. proposed the concept of limb remote ischemic conditioning (LRIC) to protect the brain from ischemic injury. This method induces several local ischemia and reperfusion of the limb through repeated, transient compression and release of the limb, which can prevent the subsequent fatal ischemia or ischemia and reperfusion injury of distant brain tissue [14]. A study has reported that LRIC can reduce inflammatory response after cerebral ischemia [11]. This inflammation not only immediately affects the infarcted tissue but also causes long-term damage in the ischemic penumbra [15]. LRIC has good clinical transformation prospects owing to its safety, noninvasiveness, simplicity, and ease of use [11]. However, the role and mechanism of LRIC in neurogenesis after an ischemic stroke are not well understood.

Evidence suggests that the Notch1 signaling pathway plays a regulatory role in neurogenesis of the subgranular zone (SGZ) and subependymal ventricular zone (SVZ) [16]. Notch1 was originally identified in Drosophila. The Notch1 signaling pathway is highly conserved and is expressed in multiple species, including invertebrates and mammals [17]. The classical Notch1 signaling pathway determines the fate of the nerve cells during brain development and regulates neural cell proliferation, differentiation, and apoptosis [18]. Notch1 signaling molecules in the SVZ are activated by focal ischemia, and inhibition of the Notch1 signaling pathway can significantly inhibit the proliferation of neural stem cells, suggesting that it is closely related to neurogenesis [19].

MicroRNAs (miRNAs) are a class of non-coding single-stranded RNA encoded by endogenous genes that are approximately 22 nucleotides in length and are widely found in body fluids such as blood and cerebrospinal fluid. miRNAs in tissue cells can enter and exit cells through exocytosis [20]. Thus, miRNAs in peripheral blood circulation depict a significant correlation with miRNA changes in brain tissue [21]. miRNAs play an important role in the pathological changes in stroke-related risk factors and in the acute and subacute stages of stroke [22,23]. Further, it has been shown that miR-9, miR-124, miR-133b, and miR-21 play important regulatory roles in nerve regeneration after ischemic stroke [22]. Studies have shown that miR-449 plays an important role in nerve regeneration during mammalian embryonic development [24]. On day 14 of embryonic development, miR-449b, which regulates brain development, is expressed in the ventricles of the neocortex and SVZ [24]. Studies in the airway epithelium have shown that miR-449b represses the Notch pathway [25]. However, it is unclear how miR-449b/Notch1 pathways are involved in stroke recovery. An understanding of miRNAs’ interaction with the Notch signaling pathway could lead to new approaches to enhancing neurogenesis after ischemic stroke. A previous study reported that LRIC treatment activated Notch intracellular domain (NICD) in the arteries surrounding the ischemic area [26]. However, it remains unclear whether the miR-449b/Notch1 signaling pathway is involved in the promotion of LRIC-induced neurogenesis following ischemic stroke.

This study aimed to investigate whether LRIC could promote neurogenesis after ischemic stroke and the role of the miR-449b/Notch1 pathway in LRIC-induced neuroprotection. Further, we aim to explore the question regarding how ischemic conditioning of the limb promotes neurogenesis in distant brain tissues.

## 2. Materials and Methods

### 2.1. Animals and Model Building and Hypoxia/Ischemic Conditioning

Male C57BL/6 mice (22–25 g) were purchased from Vital River Laboratory Animal Technology Co., Ltd. (Beijing, China). The experiments were conducted in accordance with the institutional animal ethics of Capital Medical University. Figure 1a showed the experiment flow.

For the focal cerebral ischemia mice model, a middle cerebral artery occlusion (MCAO) model was generated as previously described [27]. Briefly, mice were anesthetized with 1.5% enflurane in a 30% O_2_/68.5% N_2_O mixture. Heparinized intraluminal nylon filament was used to occlude the right MCA. The suture was removed at 60 min after inserting the filament. During surgery, a heating pad was used to maintain the rectal temperature at 37 ± 0.5 °C.

The mice were randomly divided into three groups: MCAO and LRIC, MCAO, and sham. All mice were intraperitoneally injected with 5-bromo2-deoxy-uridine (BrdU) (50 mg/kg) twice per day starting from the day after MCAO. The experimental process is shown in Figure 1A.

For mice: we performed LRIC as previously described [13]. Anesthesia was administered intraperitoneally with sodium pentobarbital (30 mg/kg). We used a LRIC instrument developed by ourselves (ZL201720524103.7). LRIC was performed on each hind limb for 3 cycles at 10 min/cycle at 280 mmHg for ten minutes per cycle and followed by ten minutes of reperfusion. The blood flow to the hindlimbs was determined using two-dimensional laser speckle imaging. Successful limb ischemia was defined as a drop in blood flow below 25% of baseline (Figure 1b). The mice were treated with LRIC at 10 min after MCAO. Subsequently, the mice underwent LRIC training once a day until they were sampled. The mice in the sham and MCAO groups received sodium pentobarbital treatment alone. LRIC treatment was not given to mice in the sham and MCAO groups.

For NE-4C cells, oxygen–glucose deprivation (OGD)/reoxygenation was used to mimic ischemia/reperfusion injuries in vivo: the culture medium was replaced with FBS-glucose-free medium, and then the plates were transferred into a hypoxic chamber and incubated at 37 °C for 24 h in a cell incubator containing 8% O_2_, 5% CO_2_ and 87% N_2_. At the same time, the normoxic group underwent normal culture conditions.

### 2.2. Two-Dimensional Laser Speckle Imaging

Two-dimensional laser speckle imaging was used to monitor cerebral blood flow (CBF) before and after cerebral ischemia (10 min after MCAO). Blood flow ratios of the left and right cerebral hemispheres were determined, and the MCAO model was considered successful when the ratio was <10% (Figure 1b,c).

### 2.3. Neurological Function Assay

Asymmetric motor behavior was tested using the elevated body swing test (EBST) (N = 10 each group) [28]. We held mice at the base of their tails and raised them 10 cm from the test surface. The rotation of the upper body to one side more than 10 degrees was recorded as one point, and each mouse was recorded for 30 trials. The number of rotations (left and right) was recorded for each mouse. The total number of swings to the left was divided by 30 (number of trials) to obtain the percentage of swings to the left. Modified neurological severity scores were used to test a composite of motor, sensorimotor and reflex integration. The test was graded on a score of 0 to 12 [29]. A higher score indicated a more severe neurological deficit. The detector was blinded to the group and treatment allocation during the neurological function tests.

### 2.4. Generation of the Notch1 shRNA Lentivirus and Lateral Ventricle Implantation

To knockdown Notch1, lentivirus vectors which expressed mouse Notch1 shRNA and scrambled shRNA were obtained from Sigma (Sigma Co., Ltd., Ronkonkoma, NY, USA). The Notch1 shRNA lentivirus was obtained as previously described [30]. The scrambled shRNA sequence (5′-TTCTCCGAACGTGTCACGT-3′) or Notch1 shRNA sequence (5′-CCGGCCCACATTCCAGAGGCATTTACTCGAGTAAATGCCTCTGGAATGTGGGTTTTT-3′) was subcloned into the pLV/H1-GFP plasmid. Recombinant lentivirus titer was adjusted to 1 × 10^9^ TU/mL.

Lentivirus (3 mL) was stereotaxically injected into the right lateral ventricle (0.7 mm outside to the midline, 0.1 mm posterior to bregma, and 2.2 mm deep). All implanted mice were operated as sham or MCAO model at day 14 after virus injection.

### 2.5. Cell Culture, miR-449b shRNA or/and Notch1 shRNA Infection

NE-4C was acquired from American Type Culture Collection (ATTC, Manassas, VA, USA). The cells were cultured in EMEM (Eagle’s Minimum Essential Medium, ATCC) containing 10% FBS (Gibco), 1% penicillin–streptomycin (Thermo Fisher, Waltham, MA, USA) and 2 mM L-glutamine (Sigma). The cells were seeded into poly-D-lysine (PDL, Sigma)-coated dishes (4 × 10^4^ cells/cm^2^ density). The medium was changed 3 times in one week. Cells were treated with miR-449b shRNA and/or Notch1 shRNA lentivirus (10^7^ TU) in a 12-well plate for 48 h.

### 2.6. Immunofluorescence Analysis

Mouse brain tissue samples were obtained on day 7 after MCAO. Freezing brain tissues were coronal sectioned into 10 μm slices as previously described [31]. For detection of BrdU-labeled cells, slices were incubated with 2 N HCl for 20 min at 37 °C, then rinsed with borate (0.1 M, pH 8.5) for 10 min. The sections were incubated with primary antibodies: rabbit anti-BrdU (1:500, GeneTex, Irvine, CA, USA) and goat anti-doublecortin (1:100, Santa Cruz, Dallas, TX, USA)

The NE-4C cells (2.5 × 10^4^/cm^2^) were cultured in a 24-well plate (Corning Costar). After incubation for 24 h (cells reached 80% confluency), the cells were washed 3 times with PBS. Cells were fixed in cold 4% PFA for 30 min and then washed with PBS. Then, the cells were permeabilized with 0.3% Triton X-100 for 15 min; following this, the cells were blocked with 1% BSA. To stain the cells, we added 30 μM BrdU 30 into the medium and cultured it for 4 h. The next staining process was the same as above. The primary antibodies were rabbit anti-BrdU (1:500, GeneTex), mouse anti-nestin (1:500; Millipore, Billerica, MA, USA), and rabbit anti-SOX2 (1:500, GeneTex).

### 2.7. Western Blotting

Western blotting was used to analyze the activation of the Notch1 signaling pathway in mice. Proteins were isolated from mouse brain microvessels on day 7 after MCAO. For other protein analysis, the protein isolated from SVZ protein extraction solution (RIPA) (Applygen Technologies Inc., Beijing, China) was used for cell lysis. A total of 20 μg protein was electrophoresed on 10% SDS-PAGE and then transferred to a PVDF membrane (Millipore Corporation). After blocking the membrane with 5% skim milk (BD Difco) in TBST for 1 h, the membranes were incubated with the primary antibody at 4 °C overnight. Image J software (National Institutes of Health, Bethesda, MD, USA) was used for the calculation of gray value. Experimental conditions were blinded to the observer. Six mice were used for each group.

The NE-4C cells were cultured in 100 mm dishes (Corning Costar, Cambridge, MA, USA). After the cells reached 70% confluence, cells were incubated under hypoxic or normal culture conditions for 24 h. Then, the cells were washed 3 times with PBS and lysed with RIPA lysis buffer (Applygen Technologies Inc., Beijing, China). The remaining procedures were the same as those described above.

The primary antibodies used were rabbit anti-NICD (1:1000; Cell Signaling Technology, Danvers, MA, USA) and rabbit anti-Notch1 (1:100; Cell Signaling Technology)

### 2.8. Real Time RT-PCR

To quantify miR-449b, total miRNA was purified using the mirVanaTM miRNA Isolution Kit (Qiagen, Gaithersburg, MD, USA), and reverse transcription was performed using the miScript II RT Kit (Qiagen). The expression of miRNA-449b was determined using the miScript SYBR Green PCR Kit (Qiagen). Primer sequence upstream of miR-449b: GGGAGGCAGTGTATTGTTAGCTG. Primer sequence upstream of U6: CGCTTCGGCAGCACATATACTA. Downstream primers were obtained using the miScript II RT Kit (Qiagen).

### 2.9. Luciferase Assay

H293T cells were co-transfected with either miR-449b mimic or a scrambled control miRNA, and luciferase constructs containing the Notch1 3′-UTR (site-directed mutant or wild-type). After 24 h transfection, the cells were harvested. The dual-luciferase reporter assay system (E1910; Promega, Madison, WI, USA) was used for the dual-luciferase assay.

### 2.10. Statistical Analysis

The sample sizes are shown in each figure. All values are expressed as mean ± SD. The statistical analyses were performed with SPSS for Windows (version 22.0; SPSS Inc., Chicago, IL, USA). Student’s t-test for two groups, or one-way ANOVA followed by Tukey’s post-hoc test for more than two groups, were performed. *p* < 0.05 was deemed significant.

## 3. Results

### 3.1. LRIC Improved Neurological Function at 28 Day after MCAO

Figure 1a shows the experimental process. We established the MCAO model and used laser speckle to monitor cerebral blood flow to determine the stability of the model (Figure 1b,c). Next, we explored whether LRIC was beneficial for neurological recovery. As shown in Figure 2, neurological impairment and motor behavior asymmetry (body elevation swing test) were significantly improved in the MCAO and LRIC group 28 days after MCAO, compared with the MCAO group (*p* < 0.05) (Figure 2).

### 3.2. LRIC Promote the Proliferation and Migration of SVZ Neural Stem Cells

To further prove the mechanism by which LRIC promotes neurological function recovery, we explored its effect on neurogenesis and investigated its effects on the proliferation and migration of neural stem cells in the SVZ on day 7 after MCAO. The results showed that the number of DCX-positive cells in the SVZ and striatum is significantly different between the MCAO and LRIC and MCAO groups (Figure 3a,b). Immunofluorescence double-standard staining showed a significant difference in the number of DCX-positive and BrdU-positive cells in the SVZ and striatum, between the groups (Figure 3a,c). These results suggest that LRIC significantly promotes the proliferation and migration of neural stem cells.

To investigate the association between the increase in the number of BrdU-positive/DCX-positive cells after LRIC treatment and the neurological function scores, Pearson product regression analysis was used which showed that, after 28 days of LRIC treatment, the neurological function score was negatively correlated with the number of DCX-positive/BrdU-positive cells in striatum and SVZ (r^2^ = 0.6120, *p* = 0.0026, r^2^ = 0.4679, *p* = 0.0142, respectively) (Figure 4). The results suggested that the increase in the number of BrdU-positive/DCX-positive cells may be an important part of the recovery of neurological function after LRIC.

### 3.3. LRIC Promoted Activated Notch1 (NICD) Protein Expression in SVZ

To test whether LRIC can activate Notch1 signaling, NICD was evaluated by Western blotting at 7 days after MCAO. The expression of NICD was significantly increased in the SVZ of MCAO mice compared with the sham group (*p* < 0.01). LRIC further increased the expression of NICD compared with the MCAO only group (*p* < 0.05) (Figure 5a).

### 3.4. Noth1 shRNA Abolished the Beneficial Effects of LRIC on Proliferation of Neuronal Stem Cells in MCAO Mice

To further demonstrate that the Notch1 signaling pathway mediates LRIC and promotes neurogenesis after ischemic stroke, we injected Notch1 shRNA virus into the mouse SVZ to knockdown Notch1 expression. Neural precursor cells (DCX labeled) that migrated to the basal ganglia were detected by immunofluorescence double-labeling 14 days after MCAO. Our results showed that the number of DCX/BrdU double-positive cells significantly increased in the MCAO and LRIC group compared with the MCAO group (*p* < 0.05); however, the number of cells in the Notch1 knockdown group was significantly decreased (*p* < 0.05) (Figure 5b,c). These results suggest that the Notch1 signaling pathway may mediate LRIC to promote neurogenesis after ischemic stroke.

### 3.5. LRIC Down-Regulated the Expression of miR-449b in SVZ

We used real-time PCR to detect the expression of miR-449b in the plasma and SVZ, and the results showed that the expression of miR-449b did not significantly change in the MCAO group compared with the sham group. However, LRIC downregulated the level of miR-449b compared with the MCAO group in both the plasma and SVZ (*p* < 0.01) (Figure 6a). Pearson product regression analysis showed that, after 28 days of LRIC treatment, the neurological function score was positively correlated with the level of miR-449b in the plasma and SVZ (r^2^ = 0.6814, *p* = 0.009, r^2^ = 0.4474, *p* = 0.0174, respectively) (Figure 6b).

### 3.6. miR-449b Was Involved in the Proliferation of Neural Stem Cells

NE-4C were transfected with miR-449b inhibitors and cultured in a medium containing EGF and FGF. Neural progenitor cells were identified using SOX2 and Nestin immunofluorescence staining. Results showed that mir-449b inhibitors significantly promoted the proliferation of neural stem cells (Figure 7).

### 3.7. miR-449b Targeted Notch1 and Modulated the Notch1 Level in Cells

Figure 8a shows that the 3’UTR of Notch 1 mRNA has potential binding sites with miR-449b. As shown by luciferase reporter experiments, miR-449b inhibits the luciferase activity of Notch1. Mutations at these binding sites significantly abolished luciferase reporter responsiveness to miR-449b (*p* < 0.01) (Figure 8b). Moreover, Western blot analysis revealed that Notch1 expression was increased by miR-449b inhibitor treatment (*p* < 0.05) (Figure 8c,d).

### 3.8. miR-449b/Notch1 Signaling Pathway Modulated NE-4C Proliferation

To test whether miR-449b targets Notch1 to modulate neural stem cell proliferation, both the miR-449b inhibitor and Notch1 shRNA were co-transfected into NE-4C cells. The miR-449b inhibitor significantly promoted NE-4C proliferation compared with the control group (*p* < 0.05). Notch1 shRNA significantly attenuated NE-4C proliferation compared with the miR-449b inhibitor group (*p* < 0.01). Both the miR-449b inhibitor and Notch1 shRNA attenuated the promotion of induced miR449b (*p* < 0.05) (Figure 9).

## 4. Discussion

LRIC has been reported to provide neuroprotection against cerebral ischemia in the past. This is the first report to study the effect of LRIC on neurogenesis following cerebral ischemia. The study is also the first to demonstrate that the miR-449b/Notch1 pathway is involved in neurogenesis after cerebral ischemia. Our novel observations were as follows: (1) LRIC upregulated the expression of NICD in the SVZ; (2) LRIC effectively decreased the level of miR-449b, and the neurological score was positively correlated with the level of miR-449b; (3) miR-449b targeted Notch1 and modulated Notch1 levels in cells; (4) miR-449b inhibition promoted the proliferation of neural stem cells, and the promoting effects of miR-449b inhibition on neurogenesis were ablated by application of Notch1 shRNA.

LRIC has an exact protective effect against cerebral ischemia [11]. For the treatment of stroke in chronic convalescence, reports have clarified that LRIC can promote the recovery of neurologic function by promoting arteriogenesis and increasing neuroglobin [26,32]. In the subacute and chronic stages of stroke, the recovery of neurological function mainly depends on brain plasticity, including regeneration and repair, such as nerve regeneration, angiogenesis, and myelin regeneration [33]. The process of neurogenesis plays a key role in neurological recovery and improving patient outcomes [34]. The literature shows that one cycle of local ischemic conditioning applied to the MCA could promote neurogenesis 2 weeks post-stroke [35]. However, due to the characteristics and properties of the brain, local ischemic conditioning has significant limitations [11,36]. This study first reported that LRIC improved neuronal outcomes through the promotion of neurogenesis over a longer follow-up period, as indicated by the Pearson association analysis which showed that the neurological function score was negatively correlated with the number of BrdU-positive/DCX-positive cells in the SVZ and striatum. In our previous study, we found that hypoxic conditioning in a hypoxic chamber could promote neurogenesis [37]. Consequently, both hypoxic postconditioning and LRIC can promote neurological function recovery by promoting neurogenesis in stroke. These methods are simple, feasible, and low cost; therefore, they are expected to be widely applied in clinical practice.

Next, we explored the question regarding how ischemic conditioning of the limb promotes neurogenesis in distant brain tissue, and what the molecular mechanism is. LRIC induces endogenous protection and is a strategy to regulate gene expression changes when cells face life and death choices under extreme conditions, mainly involving epigenetic—including microRNA (miRNA)—regulation, DNA methylation, and histone modification [38]. It has been well documented that miRNAs play an important role in the central nervous system due to their ability to cross the blood–brain barrier and regulate gene expression. The miRNA molecule belongs to a special class of non-coding single-stranded RNA molecules (ncRNAs), 18–22 nucleotides in length, that bind to mRNA by pairing and degrading to prevent protein synthesis [22]. miRNAs in peripheral blood circulation are significantly correlated with miRNA changes in brain tissue [21]. Therefore, we believe that the regulation of miRNAs may be one of the key molecular mechanisms to achieve the “peripheral protection center”.

Studies have shown that miR-449 plays an important role in neurogenesis during mammalian embryonic development [24]. On day 14 of embryonic development, miR-449 is expressed in the ventricles of the neocortex and in the SVZ, the region of neural progenitors, regulating brain development [24]. A study reported abnormal brain development, ciliogenesis, and spermatogenesis in cluster miR-34/449 knockout mice [39]. These studies suggest that miR-449 is an important regulator of stem cell proliferation and differentiation. However, whether miRNA449b is involved in neurogenesis after cerebral ischemia remains unclear. In this study, we first reported that miRNA-449b is involved in neurogenesis after ischemic stroke. We further demonstrated that LRIC downregulated the levels of miR-449b in the plasma and SVZ. Pearson’s association analysis showed that the neurological function score was positively correlated with the level of miRAN-449b in the SVZ and plasma. Further studies have shown that the inhibition of miRNA449b could promote neurogenesis. Marcet et al. reported that miR-449 targets Notch, and regulates the development of multiple ciliated cells through direct inhibition of the Delta/Notch1 signaling pathway [25]. The Notch1 pathway plays an essential role in adult neurogenesis [16]. Notch1 signaling molecules in the SVZ are activated after focal ischemia, and inhibition of the Notch1 signaling pathway can significantly inhibit the proliferation of neural stem cells [19]. In this study, we report for the first time that miR-449b/Notch1 is involved in the regulation of neurogenesis after ischemic stroke. We also found that LRIC modulated miR-449b/Notch1 to promote neurological function recovery during the chronic phase of stroke. In brief, LRIC suppresses miR-449b expression, thereby activating the Notch1 signaling pathway, promoting neurogenesis after ischemic stroke, and ultimately improving neural function.

This study had several limitations. First, neurogenesis after ischemic stroke includes the proliferation, migration, and differentiation of SVZ neural stem cells. There are four types of neural stem cells in the process of proliferation, and each type of cell has a different role, and, therefore, the cytobehavioral mechanisms mentioned above are indispensable for the whole process of neurogenesis [40]. The effect of LRIC on specific cell types and processes was not evaluated. Thus, future studies should clarify the specific steps of neurogenesis following cerebral ischemia. Second, downstream molecules of the miR-449b/Notch1 signaling pathway were not detected. We aim to examine the molecules and pharmacological inhibitors in genetically modified mouse models which could be useful in future studies. Third, neural progenitor proliferation and migration can be influenced by both pro- and anti-inflammatory cytokines [41]. In the future, it will also be possible for us to investigate whether LRIC affects neurogenesis by affecting inflammation.

## 5. Conclusions

In conclusion, this study revealed that LRIC exerts neuroprotective effects by promoting neurogenesis in MCAO mice. Furthermore, miR449b expression changed following an ischemic injury to SVZ neural stem cells. Our experiments in vitro and in vivo showed that LRIC regulates miR-449b/Notch1 to promote neurogenesis in neuronal stem cells. Taken together, LRIC promoted neurogenesis after cerebral ischemia in part by regulating the miR-449b/Notch1 pathway and, thus, plays a role in neuroprotection.

## Figures and Tables

**Figure 1 biomolecules-12-01137-f001:**
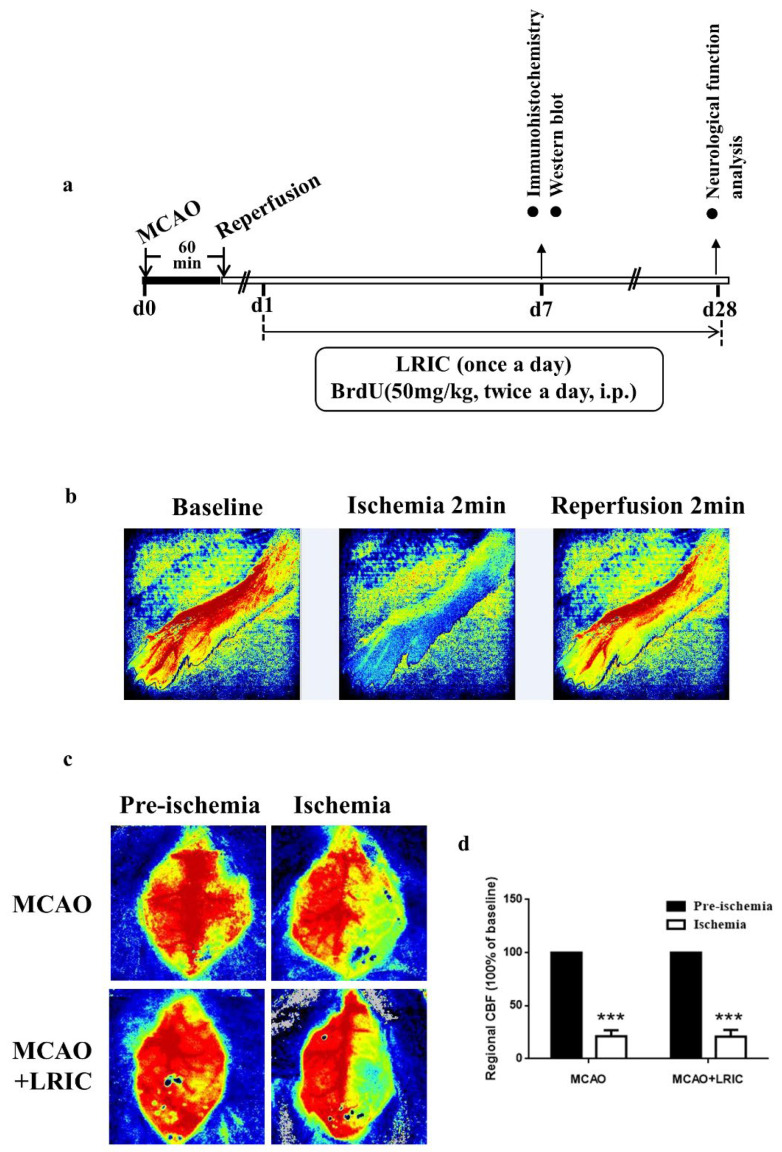
Experimental flow. (**a**) Experiment flow chart. (**b**) Representative images of blood flow in the left hind leg using a laser speckle before LRIC, 2 min after one cycle of LRIC, and 2 min after reperfusion. (**c**) Representative diagram of laser speckle. (**d**) Histogram of cerebral blood flow. *** *p* < 0.001, N = 10 per group.

**Figure 2 biomolecules-12-01137-f002:**
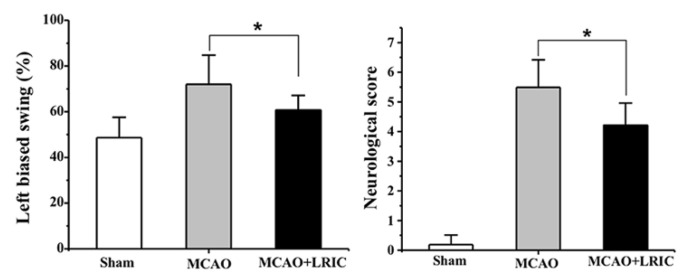
Effects of LRIC on neurobehavioral function after focal ischemia. The elevation body swing test (the higher the percentage, the more severe the defect) and the neurobehavioral scoring system were used to score the neurological impairment. * *p* < 0.05, N = 10 per group.

**Figure 3 biomolecules-12-01137-f003:**
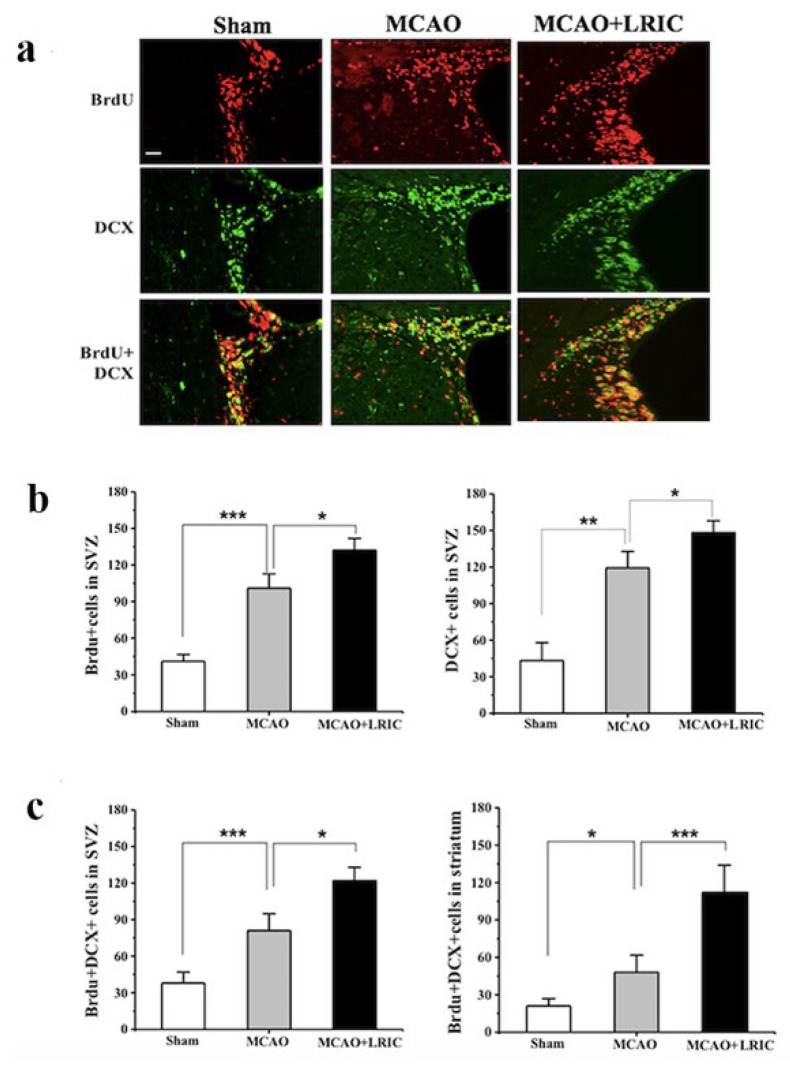
LRIC promoted the proliferation and migration of neural stem cells. (**a**) At day 7 after MCAO, 5 mm thick paraffin embedded sections were used to perform immunohistochemical staining; positive signals in the figure are DCX-positive cells. Immunofluorescence image shows the expression of BrdU (red) and DCX (green) in SVZ at 7 days. Scale bar = 50 mm. (**b**,**c**) The graph shows the number of BrdU-positive/DCX-positive cells in SVZ and striatum, respectively. Results show mean ± SD, N = 6 per group. * *p* < 0.05, ** *p* < 0.01, *** *p* < 0.001.

**Figure 4 biomolecules-12-01137-f004:**
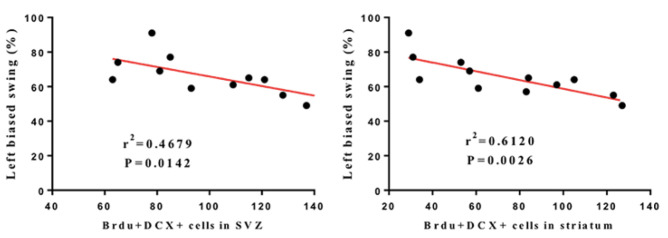
Pearson correlation between left biased swing at 28 days after MCAO and number of BrdU-positive/DCX-positive cells.

**Figure 5 biomolecules-12-01137-f005:**
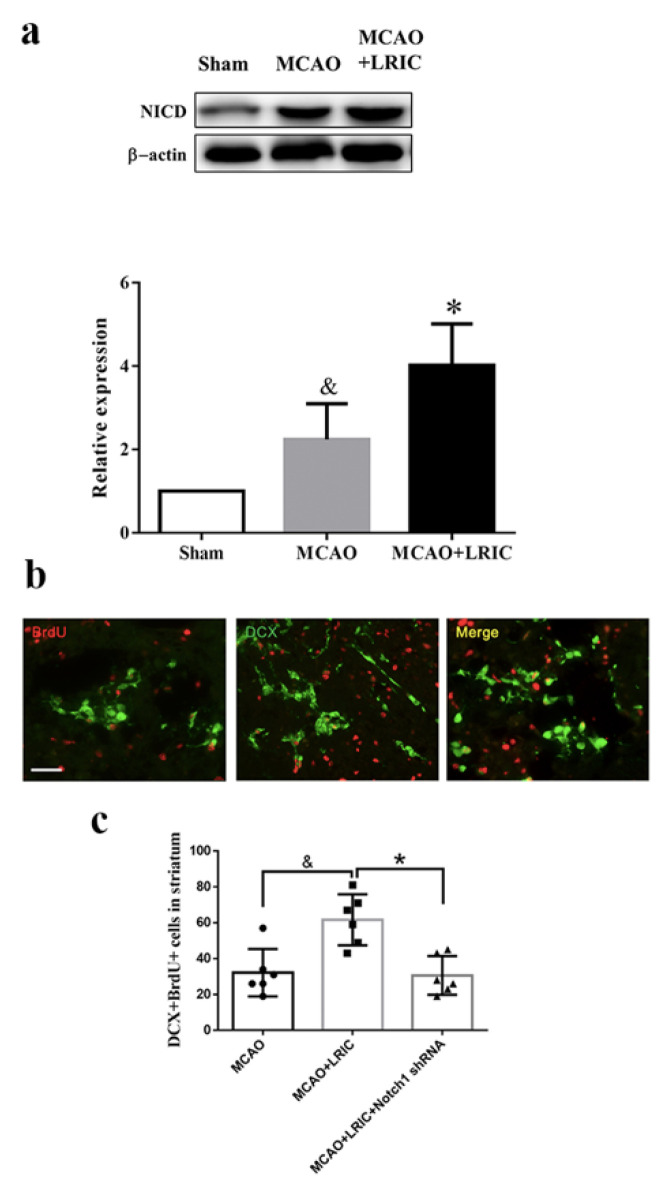
LRIC promoted activated Notch1 (NICD) protein expression in SVZ, and Noth1 shRNA abolished the beneficial effects of LRIC on proliferation of neuronal stem cells in MCAO mice. (**a**) Western blot detection of NICD expression in SVZ and showing the semi-quantitative results of protein expression in the bar graph. N = 6 per group, * *p* < 0.05. vs. MCAO group, and *p* < 0.01. vs. sham group. (**b**) Representation of immunofluorescence double staining. Scale bar = 100 mm. (**c**) Bar graph shows the number of DCX/BrdU double-positive cells. N = 6 per group. * *p* < 0.05, and &, *p* < 0.05.

**Figure 6 biomolecules-12-01137-f006:**
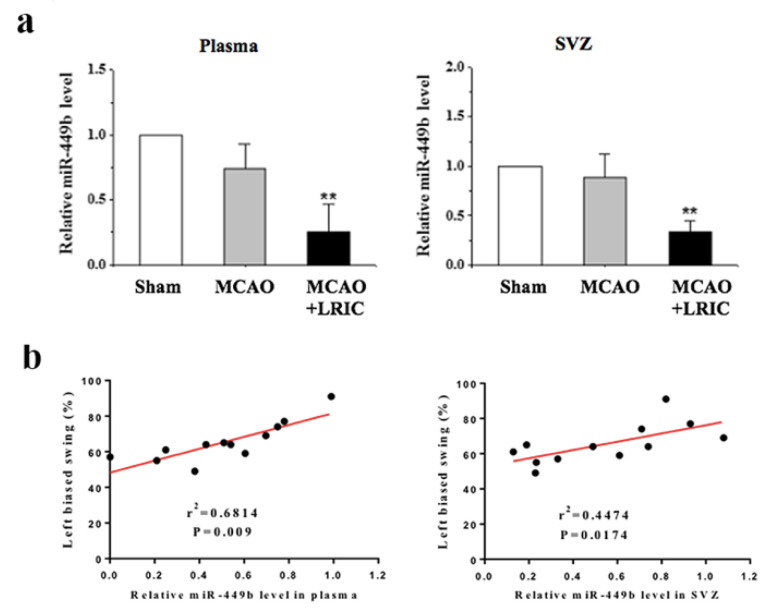
LRIC down-regulated the expression of miR-449b in plasma and SVZ. (**a**) The graph shows the level of miRNA-449b in the plasma and SVZ. N = 8/each group. ** *p* < 0.01. (**b**) Pearson correlation between left biased swing at 28 days after MCAO and number of BrdU-positive/DCX-positive cells.

**Figure 7 biomolecules-12-01137-f007:**
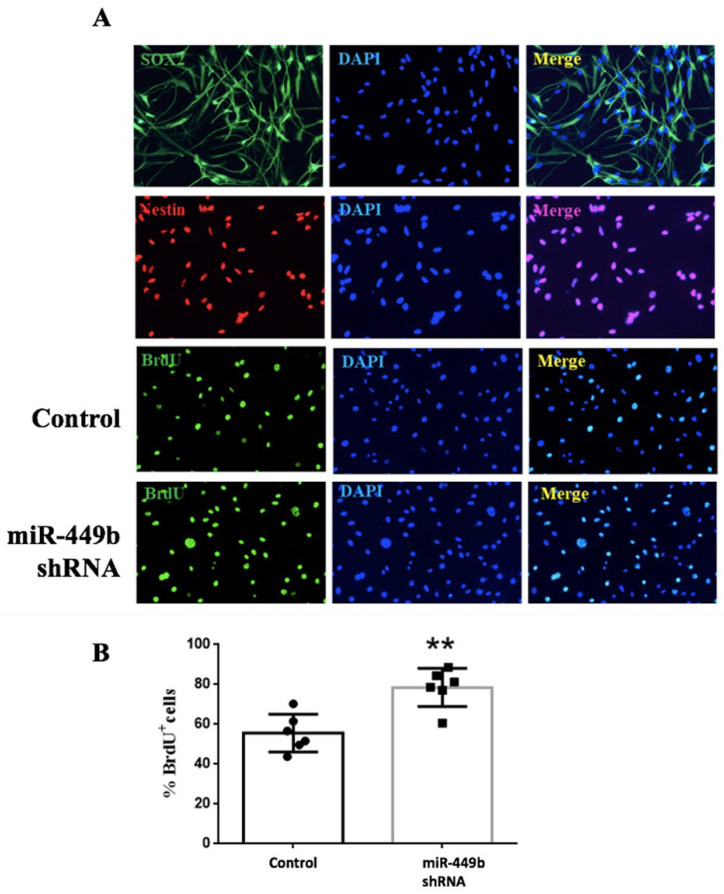
Effect of miR-449b on proliferation of neural stem cells. (**a**) Representative image of immunofluorescence staining. (**b**) Bar graph shows BrdU positive cells, and miR-449b inhibitor significantly increased the number of BrdU-positive cells, ** *p* < 0.01, N = 6 per group.

**Figure 8 biomolecules-12-01137-f008:**
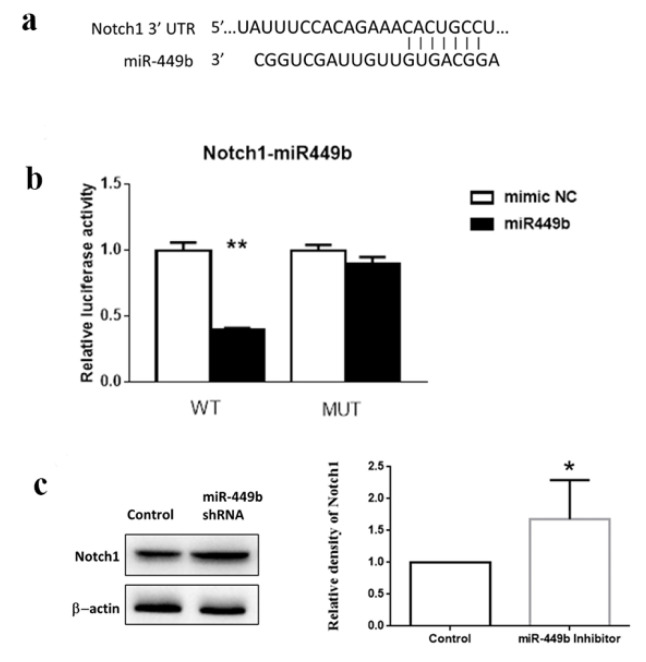
miR-449b targeted Notch1 and modulated the Notch1 level in cells. (**a**,**b**) H293T cells were co-transfected with either miR-449b mimic or scrambled control miRNA and a luciferase construct containing the Notch1-UTR (wild-type or mutant). After 48 h, cells were collected, and the dual luciferase assay was performed. ** *p* < 0.01 vs. negative control. (**c**) Western blot detection of Notch1 expression in NE-4C and showing the semi-quantitative results of protein expression in bar graph. N = 6 per group. * *p* < 0.05.

**Figure 9 biomolecules-12-01137-f009:**
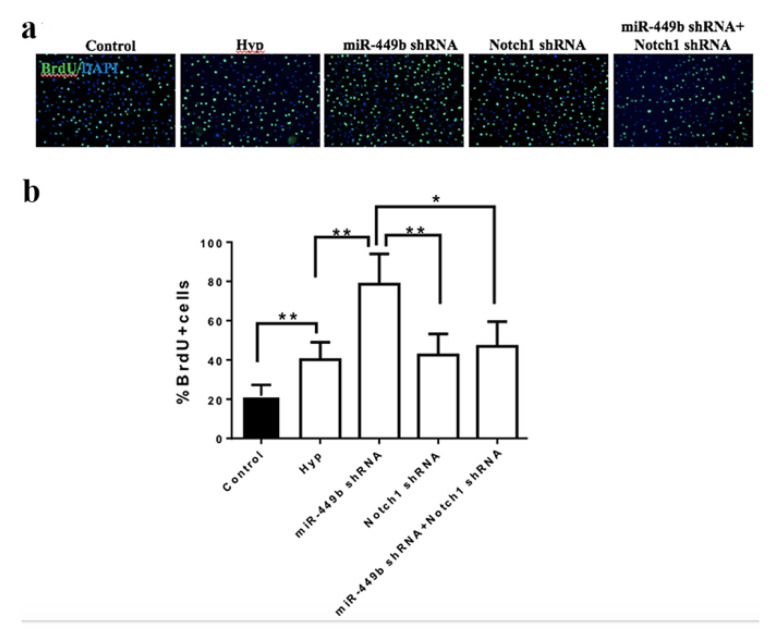
miR-449b/Notch1 signaling pathway modulated NE-4C proliferation. (**a**) Representation of immunofluorescence staining. (**b**) Bar graph shows BrdU-positive cells. N = 6 per group. * *p* < 0.05, ** *p* < 0.01.

## Data Availability

The data that support the findings of this study are available from the corresponding author upon reasonable request.

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
