# Peer review of "Limb Remote Ischemic Conditioning Promotes Neurogenesis after Cerebral Ischemia by Modulating miR-449b/Notch1 Pathway in Mice"

_biomolecules, 2022, doi:10.3390/biom12081137_

Round 1

Reviewer 1 Report

In the article “Limb remote ischemic conditioning promotes neurogenesis after cerebral ischemia by modulating miR-449b/Notch1 pathway in mice” the authors studied the influence of limb remote ischemic conditioning on neurogenesis after cerebral ischemia in mice. The investigation performed on the good level and the results of study are interesting for fundamental physiology and medicine.

The comments:

1.       The authors have written: “We performed LRIC as previously described [13].” but it isn’t clear how they estimate the level/intensity of limb ischemia.

2.       The authors have written: “LRIC treatment was not given to mice in the sham and MCAO groups.” What were manipulations performed with the sham and MCAO animals? Were the mice of sham and MCAO groups treated with sodium pentobarbital intraperitoneally (without LRIC) every day during all experiments? It is known sodium pentobarbital along can influence on ischemia results. How the repeated administration of sodium pentobarbital influences on cell proliferation?

3.       The authors have written: “For NE-4C cells: The cells were seeded in petri dishes and then incubated at 37°C  for 24 h in a cell incubator containing 8% O2, 5% CO2 and 87% N2.” Ischemia includes two main damage factors: hypoxia and glucose deprivation. What the influence of glucose deprivation on NE-4C proliferation?

4.       Lines 322 and 340: need replace “neurol” on “neural”.

Author Response

  1. The authors have written: “We performed LRIC as previously described [13].” but it isn’t clear how they estimate the level/intensity of limb ischemia.

Answers: We are sorry for this omission and added description to the methods section at line 129-131. “The blood flow to the hindlimbs was determined using two-dimensional laser speckle imaging. Successful limb ischemia is defined as a drop in blood flow below 25% of base-line (Fig. 1B).”

  1. The authors have written: “LRIC treatment was not given to mice in the sham and MCAO groups.” What were manipulations performed with the sham and MCAO animals? Were the mice of sham and MCAO groups treated with sodium pentobarbital intraperitoneally (without LRIC) every day during all experiments? It is known sodium pentobarbital along can influence on ischemia results. How the repeated administration of sodium pentobarbital influences on cell proliferation?

Answers: We apologize for our omissions in the description and added description to the methods section at line 133-134. “The mice in the sham and MCAO groups were received sodium pentobarbital treatment alone.

  1. The authors have written: “For NE-4C cells: The cells were seeded in petri dishes and then incubated at 37°C for 24 h in a cell incubator containing 8% O2, 5% CO2 and 87% N2.” Ischemia includes two main damage factors: hypoxia and glucose deprivation. What the influence of glucose deprivation on NE-4C proliferation?

Answers: We apologize for our omissions in the description and added description to the methods section at line 136-138. “Oxygen–Glucose Deprivation (OGD) /Reoxygenation is used to mimic ischemia/reperfusion injuries in vivo: the culture medium was replaced with FBS - glucose - free medium, and then the plates were transferred into hypoxic chamber……”

  1. Lines 322 and 340: need replace “neurol” on “neural”.

Answers: We have modified the errors.

Reviewer 2 Report

Authors conducted this study with the aim to investigate whether  limb remote ischemic conditioning (LRIC) was able to promote neurogenesis in cerebral ischemic  mice, and to investigate the role of the miR-449b/Notch1 pathway in LRIC-induced 28 neuroprotection. Male C57BL/6 mice (22-25g) were subjected to transient middle cerebral artery  occlusion (MCAO), and LRIC was performed in the bilateral lower limbs immediately after MCA  occlusion. Immunofluorescence staining was performed to assess neurogenesis. The cell line NE-4C was used to elucidate the proliferation of neuronal stem cells in 8% O2. After LRIC  treatment on day 28, mice recovered neurological function. Neuronal precursor proliferation was enhanced in the SVZ, and neuronal precursor migration was enhanced in the basal ganglia on day 34 . LRIC promoted the improvement of neurological function in mice on day 28, promoted neuronal 35 precursor proliferation in the SVZ, and enhanced neuronal precursor migration in the basal ganglia 36 on day 7. The neurological function score was negatively correlated with the number of 37 BrdU+/DCX+cells in the SVZ and striatum. LRIC promoted activated notch1 protein expression in 38 the SVZ and substantially downregulated miR-449b levels in the SVZ and plasma. In vitro, 39 miR-449b was found to target Notch1. Lentivirus-mediated miR-449b knockdown increased 40 Notch1 levels in NE-4C cells and increased proliferation in the cells. The effects of miR-449b 41 inhibition on neurogenesis were ablated by the application of Notch1 shRNA. Authors concluded that  that LRIC promoted the proliferation and migration of neural stem cells after MCAO, and these  effects were modulated by the miR-449b/Notch1 pathway

This is a very interesting and well conducted study

I have only minor comments to do 

Introduction

Authors should add a sentence about the role of inflammation on pathogenesis of ischemic stroke and they should add these citations on their reference section:

Siragusa S, Malato A, Saccullo G, Iorio A, Di Ianni M, Caracciolo C, Coco LL, Raso S, Santoro M, Guarneri FP, Tuttolomondo A, Pinto A, Pepe I, Casuccio A, Abbadessa V, Licata G, Battista Rini G, Mariani G, Di Fede G. Residual vein thrombosis for assessing duration of anticoagulation after unprovoked deep vein thrombosis of the lower limbs: the extended DACUS study. Am J Hematol. 2011 Nov;86(11):914-7; 

Basili S, Raparelli V, Napoleone L, Talerico G, Corazza GR, Perticone F, Sacerdoti D, Andriulli A, Licata A, Pietrangelo A, Picardi A, Raimondo G, Violi F; PRO-LIVER Collaborators. Platelet Count Does Not Predict Bleeding in Cirrhotic Patients: Results from the PRO-LIVER Study. Am J Gastroenterol. 2018 Mar;113(3):368-375; 

Pinto A, Di Raimondo D, Tuttolomondo A, Fernandez P, Arnao V, Licata G. Twenty-four hour ambulatory blood pressure monitoring to evaluate effects on blood pressure of physical activity in hypertensive patients. Clin J Sport Med. 2006 May;16(3):238-43. doi: 10.1097/00042752-200605000-00009. Erratum in: Clin J Sport Med. 2007 Mar;17(2):174.

Tuttolomondo A, Di Raimondo D, Pecoraro R, Maida C, Arnao V, Della Corte V, Simonetta I, Corpora F, Di Bona D, Maugeri R, Iacopino DG, Pinto A. Early High-dosage Atorvastatin Treatment Improved Serum Immune-inflammatory Markers and Functional Outcome in Acute Ischemic Strokes Classified as Large Artery Atherosclerotic Stroke: A Randomized Trial. Medicine (Baltimore). 2016 Mar;95(13):e3186

Discussion

Authors should a sentence underlying the role of inflammation on proliferation and migration of neuronal stem cells 

Author Response

Introduction

  1. Authors should add a sentence about the role of inflammation on pathogenesis of ischemic stroke and they should add these citations on their reference section:

Siragusa S, Malato A, Saccullo G, Iorio A, Di Ianni M, Caracciolo C, Coco LL, Raso S, Santoro M, Guarneri FP, Tuttolomondo A, Pinto A, Pepe I, Casuccio A, Abbadessa V, Licata G, Battista Rini G, Mariani G, Di Fede G. Residual vein thrombosis for assessing duration of anticoagulation after unprovoked deep vein thrombosis of the lower limbs: the extended DACUS study. Am J Hematol. 2011 Nov;86(11):914-7;

Basili S, Raparelli V, Napoleone L, Talerico G, Corazza GR, Perticone F, Sacerdoti D, Andriulli A, Licata A, Pietrangelo A, Picardi A, Raimondo G, Violi F; PRO-LIVER Collaborators. Platelet Count Does Not Predict Bleeding in Cirrhotic Patients: Results from the PRO-LIVER Study. Am J Gastroenterol. 2018 Mar;113(3):368-375;

Pinto A, Di Raimondo D, Tuttolomondo A, Fernandez P, Arnao V, Licata G. Twenty-four hour ambulatory blood pressure monitoring to evaluate effects on blood pressure of physical activity in hypertensive patients. Clin J Sport Med. 2006 May;16(3):238-43. doi: 10.1097/00042752-200605000-00009. Erratum in: Clin J Sport Med. 2007 Mar;17(2):174.

Tuttolomondo A, Di Raimondo D, Pecoraro R, Maida C, Arnao V, Della Corte V, Simonetta I, Corpora F, Di Bona D, Maugeri R, Iacopino DG, Pinto A. Early High-dosage Atorvastatin Treatment Improved Serum Immune-inflammatory Markers and Functional Outcome in Acute Ischemic Strokes Classified as Large Artery Atherosclerotic Stroke: A Randomized Trial. Medicine (Baltimore). 2016 Mar;95(13):e3186

Answers: We added description to the introduction section at line 72-74, and we added relevant references.

  1. Discussion

Authors should a sentence underlying the role of inflammation on proliferation and migration of neuronal stem cells

Answers: Thank you for your suggestion, we added the description to the discussion section at line 415-418.

This manuscript is a resubmission of an earlier submission. The following is a list of the peer review reports and author responses from that submission.